# Development of a Guideline to Enhance the Reporting of Concept Mapping Research: Study Protocol

**DOI:** 10.3390/ijerph19127273

**Published:** 2022-06-14

**Authors:** Sandesh Pantha, Martin Jones, Richard Gray

**Affiliations:** 1School of Nursing and Midwifery, La Trobe University, Bundoora, Melbourne, VIC 3086, Australia; martin.jones@unisa.edu.au (M.J.); r.gray@latrobe.edu.au (R.G.); 2Department of Rural Health, University of South Australia, Whyalla Campus, Whyalla Norrie, SA 5608, Australia; 3IIMPACT in Health, University of South Australia, Adelaide, SA 5000, Australia

**Keywords:** concept mapping, reporting guideline, methods, protocol, transparency

## Abstract

Reporting guidelines are intended to enhance the clarity and transparency of research publications. Concept mapping, a mixed-methods design, has been widely used in health research. Current reporting guidelines for mixed-methods studies are not relevant for concept mapping research. The aim of this study is to develop a reporting guideline for concept mapping research following the EQUATOR network toolkit. Guideline development is in three stages: 1. A systematic review to identify key components of reporting concept mapping research, 2. A concept mapping study involving the key stakeholder groups—researchers, methodologists, peer reviewers, journal editors, statisticians, and people who have participated in concept mapping research—to identify candidate items to include in a reporting guideline, and 3. Development of a draft reporting guideline for concept mapping research. The outcome of the research will be a reporting guideline for concept mapping research.

## 1. Introduction

Concept mapping is a type of mixed-methods study design where participants generate and prioritize ideas to develop an understanding of a complex phenomenon [1,2]. Concept mapping research has six phases—preparation, brainstorming (idea generation), structuring of the statements (prioritization and ranking), representation of statements, interpretation of maps, and utilization of the map. Concept mapping has been extensively used in public health, education, and social science research [2,3,4,5,6,7,8,9,10]. For example, Grewal et al., (2021) used concept mapping to understand barriers to self-management for people with diabetes [7]. Concept mapping has also been used to inform the development of patient-reported outcome measures such as quality of care in diabetic patients [11]. Rosas and Ridings (2017) reported that at least 23 measurement tools and scales have been developed using the concept mapping approach [12]. The number of concept mapping studies that have been published in the scientific literature has increased over the past 20 years [3,13]. Donnelly (2017) reported a 400% increase in the number of doctoral dissertations using concept mapping research between 2000 and 2014 [13].

Two systematic reviews have examined the quality of reporting concept mapping research [3,13]. Donnelly (2017) reviewed 104 concept mapping dissertations and reported frequent omission of important information across the different phases of concept mapping research [13]. Rosas and Kane (2012) reported a review of the quality and rigor of 69 concept mapping studies [3]. The review authors note that studies frequently omitted important information about concept mapping. Both reviewers conclude that there is a need for guideline to enhance the quality of reporting of concept mapping research.

A reporting guideline is a checklist of core information to include in a manuscript or research report [14]. Authors have demonstrated that reporting guidelines enhance the quality, clarity, and transparency of a research report [15,16,17,18]. For example, a review by Nawijn et al., (2019) showed that the implementation of the PRISMA (Preferred Reporting Items for Systematic Reviews and Meta-analysis) reporting guideline led to a marked improvement in the comprehensiveness of the reporting of 112 systematic reviews published in five Emergency Medical Journals.

The EQUATOR (Enhancing the QUAlity and Transparency Of health Research) network is a repository of 486 reporting guidelines covering a broad range of research methodologies [14]. To date, there are no reporting guidelines for concept mapping research listed on the EQUATOR network. However, Group Wisdom^®^,—a software package specifically designed to support concept mapping research—has suggested ten essential components to describe group concept mapping data collection and results [19]. There are several limitations to this checklist: 1. The checklist can only be applied to studies using the Group Wisdom^®^ software package, 2. It is unclear how the checklist was developed, and 3. There are important omissions in the checklist including details of how the research question was developed and reporting the number of statements generated and filtered during the process of reduction.

We identified two concept mapping studies where the authors adapted reporting guidelines from observational [20] and mixed-methods research [21]. Dopp et al. (2020) used Good Reporting of a Mixed-Methods Study (GRAMMS) to report the methodological rigor of a concept mapping manuscript [21]. The GRAMMS, however, does not address many of the key methodological components of concept mapping such as prioritization and clustering.

The aim of this study is to develop a comprehensive reporting guideline for concept mapping research adhering to the recommendations made by the EQUATOR network for reporting guideline development. Our guideline will support researchers in reporting the important parameters of concept mapping research. It may further contribute to enhancing the quality of reporting, provide transparency, and aid replicability of concept mapping research. In this protocol, we set out a detailed description of the guideline development methodology.

## 2. Methods

There are three elements to our study:A systematic review of previous concept mapping studies to identify candidate items to include in a reporting guideline.A concept mapping study to identify key items that should be included in the guidelines.Drafting of reporting guidelines for concept mapping research.

We have prospectively registered our study with 1. Open Science Framework (OSF) on 17 July 2021 (https://osf.io/nrvjh/, accessed on 24 February 2022) and 2. EQUATOR network on 29 March 2022.

### 2.1. Stage 1: Systematic Review

Our review of previous health-related concept mapping research has three aims: 1. Identify candidate items to include in the reporting guideline, 2. Identify inconsistencies in reporting, and 3. Identify the potential participants (researchers, journals, editors, and peer-reviewers previously involved in concept mapping studies) with a view to inviting them to participate in stage 2 of this project. Our review methodology adheres to the PRISMA-P statement (Appendix A) [22]. As our review does not focus on a health-reported outcome, the review is not eligible for registration with the PROSPERO (Prospective Register of Systematic Review) [23].

#### 2.1.1. Eligibility Criteria

We will include any primary concept mapping study that meets the following criteria-1. The focus of the research is a health topic (because EQUATOR guidelines only relate to this field of research) and 2. Published in a peer-reviewed journal. No restriction will be placed based on the date of publication. We will exclude studies not in English as there is evidence to suggest that the exclusion of studies in other languages has little or no influence on the strength of evidence generated from a systematic review [24,25,26].

#### 2.1.2. Information Sources

We will conduct a systematic search of three online databases: the Medical Literature Analysis and Retrieval System Online (MEDLINE), PsycInfo, and Cumulative Index to Nursing and Allied Health Literature (CINAHL). MEDLINE, PsycInfo, and CINAHL databases capture the majority of research conducted in health-related disciplines. MEDLINE and PsycInfo will be accessed via Ovid. We will access CINAHL through the EBSCOhost platform.

#### 2.1.3. Search Strategy

We followed the Peer Review of Electronic Search Strategies (PRESS) for developing the search strategy [27]. PRESS guidelines suggest developing a search strategy using the keywords (full and various truncations) and indexing terms (for example, Medical Subject Headings (MeSH)) that can then be combined using the bullion operators (OR) [28]. We adapted a search strategy previously used by Donnelly (2017) for a systematic review of concept mapping studies [13]. Our search terms include “concept map *”, “structured conceptualization”, “Ariadne”, and “concept systems”. The search strategy was initially developed for MEDLINE (Appendix A) and subsequently customized for other databases (PsycInfo and CINAHL).

#### 2.1.4. Study Records

##### Data Management

Covidence, an online systematic review management package, will be used [29]. Results from searches of each database will be exported to Endnote X9.3 as a .enl file. The citations will then be combined and exported to Covidence as a .xml file. Duplicate citations will be identified and removed in Covidence.

##### Selection Process

Screening of the articles will be carried out in two steps against our eligibility criteria (Section 2.1.1): 1. Title and abstract and 2. Full-text screening. Two reviewers will independently screen the articles. Any conflicts during the title and abstract screening will be resolved by mutual agreement. Conflicts that arise during full-text screening will be resolved after a discussion of the information available on the manuscript. If the agreement cannot be reached among the reviewers, a third researcher will provide the final decision on the inclusion or exclusion of the article.

##### Data Collection Process

We have developed a draft data extraction table (Appendix A). To test the extraction table, we will extract data for the first five studies, then meet—as a research team—and review and amend as necessary. Data will then be extracted for the rest of the included studies. Two researchers will independently undertake data extraction. Any discrepancies in data extraction will be resolved by discussion of information available in the manuscript.

#### 2.1.5. Data Items

We will extract the following from included studies: 1. Citation (surname and initial of corresponding author, title of the manuscript, year of publication), 2. Corresponding author, 3. Detail of information reported at each phase of concept mapping, and 4. Participant characteristics.

In addition, we will extract the following information that will be used in phase two: 1. Email address for the corresponding author, 2. Names (last name and initial) of all authors 2. Title of the journal, 3. Name and email address of handling editor, 4. Name and email address of the reviewer (if available).

#### 2.1.6. Risk of Bias

Our primary aim is to identify the pattern in reporting concept mapping manuscripts. Therefore, we do not intend to undertake a risk of bias assessment.

#### 2.1.7. Stage 1—Summary

In summary, our systematic review will identify how existing concept mapping studies are reported, identify potential candidate reporting items, and gather information to inform stage 2 of this study.

### 2.2. Stage 2: Concept Mapping

The second stage of guideline development will be a concept mapping study. We acknowledge possible confusion to readers that we are using concept mapping methodology to develop a reporting guideline for concept mapping. Concept mapping is potentially an appropriate methodology for generating and prioritizing candidate items for a reporting guideline [30]. Moher et al., (2010)—describing the EQUATOR network methodology for guideline development—recommends using the Delphi methodology for identifying candidate items to include in the guideline [28]. That said, we argue that concept mapping has several important advantages over the Delphi methodology: 1. Participants generate candidate items to include in the guideline and 2. The relative importance of candidate items can be ranked (meaning that potentially less important items can be removed).

We will follow the six phases of a conventional concept mapping study as described by Kane and Trochim (2007) [30].

#### 2.2.1. Phase 1: Preparation

The first phase of concept mapping is to determine the focus of the study and decide who the participants will be. The research team has identified the focus as “What are the items that need to be included in a reporting guideline for concept mapping research”?

##### Participants

Consistent with other guideline development research [31,32,33,34,35,36,37,38], participants in our concept mapping study will be: 1. authors of concept mapping studies (researchers), 2. people involved in at least five concept mapping studies (that we consider concept mapping or methodological experts), 3. journal editors (that have published concept mapping research), 4. academic peer-reviewers (that have reviewed at least one concept mapping study), 5. Statisticians (who have conducted a statistical analysis of concept mapping research), and 6. people that have previously participated in a concept mapping study (consumers).

Study participants have not traditionally been involved in guideline development; we think this is an important omission. By engaging people that have previously participated in concept mapping, we consider that we are extending the methodological rigor of the process of guideline development.

##### Sample Size

There is no specific number of participants recommended for a concept mapping study. Kane & Trochim (2007) suggest at least five participants can produce meaningful data [30]. We reviewed eight manuscripts detailing the development of reporting guidelines for health research. Between 12 and 133 people (researchers, methodological experts, statisticians, and journal editors) participated in the guideline development process [31,32,33,34,35,36,37,38]. Six out of eight studies included between 25 and 33 participants [32,34,35,36,37,38]. The EQUATOR-network methodology (for guideline development) states that at least 30 participants—that should include researchers, journal editors, academic reviewers, methodological experts, and statisticians—contribute to the development of a guideline [39]. We, therefore, plan to recruit approximately 37 participants to the concept mapping phase of guideline development.

We will recruit the following number of participants from different stakeholder groups: 1. Researchers (that have previously undertaken a concept mapping study) (*n* = 15), 2. Journal editors (*n* = 5), 3. Academic reviewers (*n* = 5), 4. Methodologists (*n* = 5), 5. Consumers (people who have participated in a concept mapping study) (*n* = 5), and 6. Statisticians (*n* = 2).

##### Demographic Information

Participant demographic information will be collected using REDCap research data capture software prior to the brainstorming task (phase 2). We will collect the following demographic information: 1. Age (in years), 2. Gender (coded as male, female, other), 3. Country of current residence, 4. Highest academic qualification (coded as no degree, baccalaureate, post-graduate, doctoral), and 5. Stakeholder group (coded researchers, methodological experts, journal editors, peer-reviewers, statisticians, consumers).

#### 2.2.2. Phase 2: Brainstorming

Individual interviews will be conducted remotely using a password-protected videoconferencing platform. Participants will be asked to respond to the focus question “What are the items that need to be included in the report of a concept mapping study?”. Interviews will be audio recorded and transcribed.

From interview transcripts, we will extract a list of statements that are singular, specific, and not overlapping. Duplicate statements will be removed. If we generate more than 98 statements (the maximum number of statements that can be handled by the software package), we will combine similar statements until the total number is below this threshold, this is a standard part of concept mapping methodology [40,41].

#### 2.2.3. Phase 3: Structuring of Statements

We will use the online software package “Ariadne” to complete this task [40,41]. We obtained written permission from the software developer to use “Ariadne” in our research. In this phase, participants will be given a pack of statements and asked to undertake two tasks: prioritization and clustering.

##### Prioritization

In this task, statements are ranked in order of importance ((1 = least important) through (5 = most important)). Participants are instructed to ensure that each point on the scale has an equal number of statements (for example, if there are 50 statements, participants would need to place 10 statements on each point on the scale).

##### Clustering

In this task, participants are told to sort an identical list of statements into clusters (groups of statements that seem to the participant to belong together). They are then asked to generate potential labels for these clusters.

##### Instructions for Completing the Tasks

A link to the “Ariadne” website will be emailed to participants where they can undertake the prioritizing and clustering tasks. A step-by-step guide on how to complete this task (DOI: 10.26181/20037188) will be attached to the email.

Once the prioritizing and clustering tasks have been completed, we will ask participants if they would like to take part in the final interpretation phase (phase 5) of the study.

#### 2.2.4. Phase 4: Representation of Statements

Data analysis will be undertaken using the “Ariadne” software package [40]. “Ariadne” produces a binary coding for the statements based on how they are grouped. Each pair of statements grouped together will be assigned “1” and “0” will be assigned for pairs that are not grouped together. A principal component analysis is computed to generate a two-dimensional concept map. Individual statements will be represented on the map as dots. The distance between two statements is proportionate to the frequency of them being grouped together by the participants. A hierarchical cluster analysis will then be conducted to produce candidate cluster solutions. “Ariadne” will generate up to 16 separate cluster solutions, each of which needs to be reviewed and interpreted [41].

#### 2.2.5. Phase 5: Interpretation of Map

Research participants will be engaged in the interpretation of the candidate maps. This phase of the study will be conducted using a video conferencing platform. Participants who agree to be a part of the interpretation workshop will be emailed, two weeks in advance, to determine if they will be available to attend. We will send a meeting link to the first six participants who reply to our email request. We will acknowledge all other participants, whom we will not be able to accommodate in the interpretation workshop.

The research team will facilitate a discussion about the strengths and limitations of each cluster solution produced during the data analysis phase. We will then ask the participants to review and interpret each of the candidate concept maps with a view to inform the structure and items to include in the reporting guidelines. The group will select the map that, in their view, offers the most robust way of understanding the data.

#### 2.2.6. Phase 6: Utilization of Concept Maps

Statement and clusters from our concept map will be considered as candidate items (and groups) for inclusion in our reporting guideline for concept mapping research.

### 2.3. Stage 3: Development of a Draft Reporting Guideline

We will synthesize findings from phase 1 (systematic review) and phase 2 (concept mapping study) to produce a draft reporting guideline. The concept map will be elaborated using information obtained during the brainstorming sessions and how these items have been reported in existing concept mapping studies (based on the findings from the systematic review—phase 1). For this phase of the study, we will form an expert panel comprising the research team plus three concept mapping experts. We will ask concept mapping experts that participated in the study if they would like to join the expert panel. A draft reporting guideline will then be developed. The expert panel will review the preliminary draft of the reporting guideline. Feedback will be incorporated into a final draft of the reporting items for concept mapping research. The guideline will be submitted to the EQUATOR network for consideration for inclusion.

## 3. Conclusions

The outcome of the study will be a draft reporting guideline for concept mapping research. By developing a reporting guideline, we seek to enhance the rigor and transparency of reporting of concept mapping research. Reporting guidelines will help to benefit populations that are the focus of the research. Prospective registration of our study methodology with the OSF and the website of the EQUATOR network will contribute to enhancing the transparency of this guideline development process. To facilitate the use of this guideline by researchers across the globe, the guideline will be published in an open-access journal and on the website of the EQUATOR network.

By using concept mapping to inform guidelines development, we are testing a different methodology that is traditionally used (Delphi) in guidelines development and we consider that it will be informative to other research groups developing reporting guidelines.

## Data Availability

Not applicable.

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
