# Peer review of "Development of a Guideline to Enhance the Reporting of Concept Mapping Research: Study Protocol"

_ijerph, 2022, doi:10.3390/ijerph19127273_

Round 1

Reviewer 1 Report

Concepts such as clarity, transparency and consistency that stand out in the study are relevant and inherent characteristics that should be a premise in the writing of any scientific work. Any scientific advance that is presented in this sense would seem to me to be relevant. 

However, I find no clear results or evidence in the article to show that the model they present is corroborated as more effective than others. The writing of the paper seems to me to be more a description of a model that the authors present without much scientific weight.  

While the style of writing is scientific, the background and arguments do not seem so to me, as the authors spend a lot of time describing obvious aspects and lack the description of fundamental elements, as I will point out in some of the parts. 

I think the authors have enough resources to enrich the research.

Title. I think it should be completed with: the improvement brought about by the development of the reporting guideline or the methodology used, for example. It seems too simple to me.

Abstract. The wording of the objectives is not clear to me. I would use the infinitive in the enumeration indicated by the authors. After reading the abstract, as a reader, I interpret that what will be described will only be the guidelines of their model, when it is much more.

Introduction. In lines 51 - 54, aspects that the authors miss and conclusions are written. I do not think these elements should appear in the introduction. 

On the other hand, in the following paragraph, the "inconsistencies" found in two systematic reviews are correctly justified bibliographically. Although I think this approach is appropriate, I think it would be worthwhile to increase the number of reviews with other studies. 

In lines 71-72, again, a conclusion is presented that should not appear in the introductory section. Likewise with the last paragraph of this section. 

Methodology. The methodology section starts with the reiteration of ideas without arguments to reinforce them. 

If the sentences in lines 85-89 are the objectives, these should be written more clearly (use of infinitives).

In line 91, I am missing the explanation of why health related disciplines are chosen. As in line 102, the reason why publications in English, or in lines 105-109, the sources described.

Are the references to Donnelly (2017), on line 115; Moher et al., (2010); and Kane & T. (2007), on line 174, missing? At least the corresponding numbers, yes. 

What are the other databases referred to by the authors on line 119?

Regarding the Participants and Sample size, I am missing: the description of the groups, as well as why the groups were chosen and how they were distributed. 

In the Brainstorming section, what is the sample size of the individual interviews?

Conclusions. This section should certainly be expanded. Hardly anything conclusive is added to the study, but rather the ideas that have already been mentioned are emphasised. 

Bibliographical references. Could references 1, 19 and 36 not be combined?

Do references 39 and 40 refer to the same author?

Reviewer 2 Report

  • There is a lot of grammatical and typo error such as in abstract intro and all other sections of menuscript. They are needed to improve thoroughly.
  • Contribution and novelty section is missing in the end of introduction section. Add your contribution and novelty statement in the second last paragraph of your menu script. Paper organization section is missing in the end of introduction of section. Briefly describe the section and subsection of your whole menu script in one paragraph. Add this paragraph in the end of the introduction section.
  • However, the most important thing is that reviewer is not able to understand the novelty of the research as well as contribution of the author. There are a lot of already published mapping studies which are highly effective and good in term of mapping research. Then why author consider this research.
  • The literature review section is missing. In this type of research literature review is highly important in order to improve the worth of this article. Author should present a table of comparison of previous published mapping articles with this manuscript and show the importance of this menuscript why it should be consider for publication although there are already many good published mapping studies.
  • In methodology section there is no screening and selection process defined. 2 or 3 lines are not enough for screening and selection process. Prove through a systematic diagram. There is no prisma diagram and checklist of selected articles.
  • A subsection “eligibility criteria” is enough to analyze the eligibility of the menuscript.
  • In search strategy section author has not presented any strategy. What is the strategy of the research? Defined search string in order to fetch the articles from different databases.
  • However, there are also many other online databases such as MDPI, IEEE, SPRINGER, ELSVIER, why author consider only these databases?
  • The article is not presented comprehensively. An in depth review is vital for such kind of research.
  • No novelty, no contribution, not in-depth, not presented comprehensively, followed methodology is wrong, quality assessment criteria is not defined.
  • There is no concluding remarks in conclusion section. Besides, conclusion is presented very short and bookish style.
  • So I would highly recommend this article to reject. As this is not able to publish in this prestigious journal.

Reviewer 3 Report

In my opinion, the authors have presented a well-written manuscript on the reporting guideline for concept mapping research. Although the topic is very relevant in the field of health research, I believe that in order for it to be accepted in this journal, it should show the results of Stage 2 and Stage 3, and not just their design. I encourage authors to resubmit the manuscript when they are able to present their findings. Also, as a minor change, I suggest revising how it should be cited in the journal (lines 57, 59 and onwards).

Reviewer 4 Report

Some comments are suggested:

  • The introduction is very well structured. It is only recommended that a definition be provided that is as clear as possible and the most used of what a mapping study is to take into account where it starts from, although phase 2 consists of defining it in a more specific way.
  • The registration of PROSPERO or the intention to do so should be indicated in the systematic review.

Reviewer 5 Report

Would it be interesting to consider what might occur between concepts and therefore also what a mapping might not detect or cover?

Do you need to discuss relations between man and machine thoroughly in your presentation?  It might create some new perspectives and nuances

Round 2

Reviewer 2 Report

This was a revised version of manuscript. Reviewer has noticed that author has not made any changes as per reviewer suggestions. In fact author has not addressed a single review questions. The article is in highly pity form, I would strongly recommend to reject this article for publication.

Author Response

We made extensive and detailed revisions to the manuscript based on feedback from five reviewers. We do not know what is meant by the phrase "...the article is in highly pity form..." but seems to us to be a potentially derogatory and unprofessional comment. We will raise our concerns about this feedback in correspondence to the editor.  

Reviewer 3 Report

After modifications made at the request of other reviewers, it could be accepted.

Author Response

No amendments are required.